# Airborne mercury species at the Råö background monitoring site in Sweden: Distribution of mercury as an effect of long range transport

Ingvar Wängberg[1], Michelle G. Nerentorp Mastromonaco[2], John Munthe[1], Katarina Gårdfeldt[2]

[1]IVL Swedish Environmental Research Institute, Gothenburg, 41133, Sweden

[2]Department of Chemistry and Chemical engineering, Chalmers University of Technology, Gothenburg, 41296, Sweden

*Correspondence to*: Ingvar Wängberg (ingvar.wangberg@ivl.se)

**Abstract.** Within the EU-funded project, Global Mercury Observation System (GMOS) airborne mercury has been monitored at the background Råö measurement site on the west coast of Sweden from mid May 2012 to the beginning of July 2013 and from the beginning of February 2014 to the end of May 2015. The following mercury species/fractions were

measured: Gaseous Elemental mercury (GEM), Particulate bound mercury (PBM) and Gaseous Oxidised mercury (GOM) using the Tekran measurement system. The mercury concentrations measured at the Råö site were found to be low in comparison to other comparable European measurement sites. A back trajectory analysis to study the origin of air masses reaching the Råö site was performed. Due to the remote location of the Råö measurement station it receives background air about 60 % of the time. However, elevated mercury concentrations arriving with air masses coming from the south-east are

noticeable. GEM and PBM concentrations show a clear annual variation with the highest values occurring during winter, whereas the highest concentrations of GOM was obtained in spring and summer. An evaluation of the diurnal pattern of GOM, with peak concentrations in midday or early afternoon, which often is observed at remote places, shows that it is likely to be driven by local meteorology in a similar way as ozone. Evidence for that a significant part of the GOM measured at the Råö site has been formed in free tropospheric air is presented.

Keywords: GMOS; Gaseous elemental mercury (GEM); Particulate bound mercury (PBM); Gaseous oxidised mercury (GOM); Mercury oxidation in the troposphere.

## 1 Introduction

Mercury (Hg) is a poisonous metal which occurs in the Earth's crust at low concentrations, primarily in the form of the mineral cinnabar (HgS, i.e. mercury sulphide). The metal is unique in many ways as it has a lower melting point (-39°C) and a lower boiling point (357°C) than any other metal. This means that mercury is volatile at room temperature and may occur as Gaseous Elemental Mercury (GEM), i.e. as Hg atoms, in the atmosphere. Other than the noble gases, mercury is the only element which occurs as an atomic gas at normal temperatures. Airborne mercury largely consists of GEM (> 98 percent)

and small quantities of Gaseous Oxidised Mercury (GOM), i.e. divalent gaseous mercury species, as well as Particulate Bound Mercury (PBM). Mercury atoms are relatively stable, which means that they have a long residence time in the atmosphere (about a year, Schroeder and Munthe, 1998) and can be dispersed globally before they are oxidised and leave the atmosphere via wet or dry deposition. Atmospheric mercury rarely constitutes a direct risk to human health. The principal environmental concern regarding mercury is its ability to be converted to methylmercury which occurs by natural biological processes. Methylmercury is highly toxic and can bio-concentrate a million-fold in the food chain, which occurs frequently in marine and freshwater ecosystems. The latter has resulted in regulatory fish consumption guidelines and health advisories in Scandinavia and North America (Schroeder and Munthe, 1998). Atmospheric mercury has both natural and anthropogenic sources and is believed to have increased by a factor of 3 or more since industrialisation (Lamborg et al., 2002). The principal manmade emission sources of mercury are artisanal and small-scale gold production, coal combustion, non-ferrous and other primary metal production as well as cement production (AMAP/UNEP, 2015; Pacyna, et al., 2016). Re-emission of mercury to the atmosphere is caused by reduction of deposited oxidised mercury on sea and land surfaces and enhances distribution and atmospheric exposure of mercury to the environment. Mercury, re-emitted to the atmosphere from sea and land surfaces, is assumed to be equal or even much larger than the total direct anthropogenic emissions (AMAP/UNEP 2013). The largest natural emission sources of mercury are active volcanoes and weathering of bedrock and their contributions to the atmospheric burden of mercury are estimated to be less than 15 - 30 % of the manmade emissions (AMAP/UNEP 2013).

As a subtask within the project Global Mercury Observation System (GMOS), funded by the EU Framework Program (FP7) for research, technological development and demonstration, the airborne mercury species, GEM, GOM and PBM were measured at the Råö measurement station situated on the west coast of Sweden. The Tekran speciation method was used to obtain values of high temporal resolution. The GOM and PBM measurements are operationally defined, and may depend on the measurement method used. It has been argued that especially GOM is subject to interferences from ozone, water vapour etc. when measured with the Tekran system (Jaffe, et al., 2014). The chemical composition of GOM and mercury species on particles still remains an unresolved issue, which hampers development of better measurement techniques (Gustin et al., 2013; Gustin et al., 2015). At the present the only available automatic method for measuring GOM and PBM is the Tekran system, however.

Here an evaluation of continuous measurements of GEM, GOM and PBM during 2012-05-15 to 2013-07-03 and 2014-02-01 to 2015-04-30 is presented and compared to the results from measurements made at other northern European sites using the same measurement method. The result of an evaluation of the origin of air masses reaching the measurement site is also presented and shows that most of the air entering the Råö site is of background origin. Evidence for that the highest concentrations of GOM measured at the Råö site arrived with air masses from the north and to the most part are due to oxidation of GEM in the free troposphere is presented. An alternative interpretation of the diurnal pattern of GOM, with

peak concentrations in midday or early afternoon is presented which suggest that the variation in GOM is likely to be caused by a local meteorological phenomena rather than local GOM formation within the boundary layer.

## 2 Experimental Section

### 2.1 Sampling site

The Råö measurement station is a costal site on the West Coast of Sweden, about 50 km south of Gothenburg ($57^o23$'$37.76$ N, $11^o54$'$50.73$ E, 7 m above sea level) and is one of the GMOS master sites. The site is situated far away from major sources of air pollutants and the measurements are performed in background air most of the time. The annual average temperature, humidity and wind speed (± 1 standard deviation) at the site were $9 \pm 7$ $^o$C, $76 \pm 12$ % and $6 \pm 4$ m s$^{-1}$, respectively, based on measurements during 2012 to 2015. The corresponding 25[th] and 75[th] percentiles were 4, 14 $^o$C; 69, 85 % and 4, 9 m s$^{-1}$, respectively. The predominantly wind direction at the site is south west. During 2012 to 2014 the average annual precipitation amount was $720 \pm 120$ mm at the Råö site.

### 2.2 Methods

#### 2.2.1 Fractionation of mercury species in air

Mercury species in air were measured using the Tekran mercury speciation system, which constitutes of the Tekran 1130/1135 sampling modules in combination with the Tekran 1130 pump and control module and the mercury detector Tekran 2537B a Cold Vapour Atomic Fluorescence Spectrometer (CVAFS) (Landis et al. 2002, Lindberg et al., 2002), see Figure 1a and 1b. The Tekran mercury detector measures the amount of elemental mercury vapour trapped on one of two alternating gold traps after thermal desorption and fluorescence detection at 253.7 nm.

With the Tekran system GEM, GOM and PBM were simultaneously measured. The sample air is pulled through an inlet impactor separating particles large than 2.5 µm at a flow rate of 10 litres per min. GOM is adsorbed on a KCl coated quartz glass denuder adjacent to the inlet. PBM is collected on a quartz glass fibre filter upstream the denuder, while GEM is passing through the system towards the 1130 pump module. Before the pump module the airflow is divided so that 1 litre per minute is pumped to the Tekran CVAFS detector and the rest goes to the pump module and is not analysed. While GOM and PBM are collected in the 1130/1135 modules GEM values with a time resolution of 5 minutes were obtained. Within the present measurements GOM and PBM were collected during 3 h followed by an analysis cycle of 60 min. During the analysis no GEM measurement is done. Every four hour, three-hour average PBM and GOM values are obtained together with 5 min average GEM. Hence, all mercury values are obtained with a time coverage of 75 %. All mercury concentrations presented are normalised to air volumes at 273 K and 1 atm. A description on the Tekran speciation system can be found on the home page of Tekran Instruments Corporation, http://www.tekran.com/files/facts_1135_r102.pdf.

The average detection limit of GEM was found to be < 0.02 ng m$^{-3}$ based on calculations of 3 $\times$ standard deviation of GEM blanks using zero air values obtained prior to each PBM and GOM analysis cycle. No reproducibility test was performed but the reproducibility is typically 12 % according to Weigelt et al., 2013. The instrument was set to automatically calibrate every 73 h using the internal permeation source. The accuracy of the internal permeation source was tested every 6 months

against manual Hg vapour injections using a laboratory built thermostat-controlled mercury vapour source, following the CEN calibration procedure (Brown et al. 2006). Since the two calibration methods agreed within 5 % no adjustments of the internal mercury calibration source was performed during the present measurements. Since the concentrations of GOM and PBM are very low at the Råö site large effort were made to limit blanks in the measurement system. The analysis of PBM and GOM occurred during three, 5-minutes long analysis cycles. Once the blanks are at the appropriate level, PBM and

GOM always were detected during their first analysis cycles. In the following 2 cycles the Hg concentration were most of the time equal to zero. For both PBM and GOM the results from the last analysis cycles, when both the PBM trap and the denuder were at high temperature, were considered to represent blank values. Three times the standard deviation of blanks using all data yielded an average detection limit of 0.11 pg m$^{-3}$ for PBM and an average detection limit of 0.23 pg m$^{-3}$ for GOM. However, as reported by Swatzendruber et al., 2009 and Slemr et al., 2016 the Tekran instrument tend to

underestimate low concentrations and the authors suggest that a recalculation procedure should be performed. According to these findings especially the low PBM and GOM values and the reported detection limits are erroneous. In lack of the necessary data, i.e. fluorescence signal versus time from each analysis, no recalculation was performed. This means that GOM and PBM values, lower than 2 pg m$^{-3}$ are likely to be underestimated. Evaluation of measurement result and maintenance of the Tekran instruments were made according to the standard operational procedure developed within the

GMOS project (http://www.gmos.eu/) that is harmonised with the operational procedures used in the USA and Canada (Olson and Rhodes, 2011; Steffen et al., 2012). Average and median values of GEM, PBM and GOM were calculated using all data including PBM and GOM zero values. In the evaluation of the weather condition during each measurement day Lamb Weather Types and precipitation data were used.

## 2.3 Back trajectory evaluation

The Hybrid Single-Particle Lagrangian Integrated Trajectory (HYSPLIT) 4.0 model (Draxler 2003) was used to produce back-trajectories starting at the Råö site. The HYSPLIT program was run with meteorological analysed fields from the NCEP/NCAR reanalysis project a joint effort between the National Centres for Environmental Prediction (NCEP) and the National Centre for Atmospheric Research (NCAR). The input meteorological data had 2.5 degrees horizontal resolution, with 18 vertical pressure levels extending between 1 hPa and 1000 hPa. For the purpose of this study trajectories were

released at 5 different vertical levels (10, 50, 100, 250 and 500 m) every four hours to get a good representation of the transport pathways into the target region. From each of the points a trajectory was started and run backwards in time for 72 h (3 days). Six back trajectories were calculated for every day to match each of the six daily analysis cycles of GOM and PBM. To evaluate the mercury concentration as function of origin of air masses the area around Råö was divided into 16,

22.5 degrees wide sectors. There are no known emission sources that are large enough to influence the mercury concentrations at the measurement site within a circle of 150 km. Geographical points on each trajectory associated with the 16 sectors outside the 150 km circle were identified by help of a specially developed Excel Visual Basic for Applications (VBA) software. Evaluation of back trajectories showed that no significant difference could be seen between back

trajectories starting at 100 m and at 250 m, also the other starting heights gave very similar results. Those at 250 m were chosen in the further evaluation. With this computer program the involved sectors could be associated with each individual mercury measurement. Altogether the origin of air masses associated to 4193 mercury speciation measurements were evaluated.

## 3 Results and Discussion

### 3.1 Measured values

Medians and arithmetic means of the mercury concentrations measured at Råö during the period 15/5 2012 to 29/4 2015 are presented in Table 1 and are compared to mercury values obtained at other Northern European sites. The variation of the average data is indicated as ± 1 standard deviation. Mace Head is a Global Atmospheric Watch (GAW) station on the west coast of Ireland. Aucgencort Moss and Harwell are rural background inland sites in Southern Scotland and Oxfordshire,

England, respectively, whereas Waldhof is a rural inland background site situated 100 km south east of Hamburg in Germany. These measurements were performed using the same Tekran instruments as used in the present work. The measurements in Aucgencort Moss, Harwell and Waldhof were made some years earlier, and are therefore not directly comparable to the present measurements. As seen in Table 1 the TGM concentration at Mace Head is somewhat higher than at Råö. TGM measurements have been performed on these two stations on a regular basis since 1996 and the tendency of

higher values on Mace Head appears to be consistent during the entirely measurement period. The reason for this is not fully understood, but the higher TGM values at Mace Head have been attributed to mercury evasion from the North Atlantic Ocean (Wängberg et al., 2007 and references therein). The TGM/GEM values at Aucgencort Moss and Harwell are somewhat lower and are more similar to the Råö measurements. Regarding GOM and PBM, only data from Aucgencort Moss and Waldhof are available for comparison. The GOM concentration at Aucgencort Moss appears to be lower than at

Råö whereas the PBM concentration is about the same. The highest concentrations of mercury species are found at Waldhof in Germany. As is discussed below this is probably because the Waldorf site is relatively close to mercury emission sources in East Europe.

Monthly average GEM, PBM and GOM are presented as whisker and box plots in Figure 2. The GEM data is evenly

distributed within a narrow range of 1.4 – 1.5 ng m$^{-3}$ with occasional elevated values reaching above 2.5 ng m$^{-3}$. The variation in concentration of PBM and GOM is much greater and varies from under the detection limit to 56 pg m$^{-3}$ for PBM

and 26 pg m$^{-3}$ for GOM. Close to 50 % of the GOM values were under the detection limit, whereas 98 % of all PBM values were higher than the detection limit.

### 3.2 Origin of air masses reaching the Råö site

Results of the back trajectory analysis are shown in Figure 3a, b, c and d. About one third of the air masses entering the Råö site during the investigated period arrived via the sectors 202.5$^{o}$ to 270$^{o}$ as shown in Figure 3a. The GEM concentration as function of origin of air masses is almost evenly distributed in the sectors 0$^{o}$ to 90$^{o}$ and 247.5$^{0}$ to 360$^{0}$ with an average value of 1.36 ng m$^{-3}$ as shown in Figure 3b. The highest GEM values are associated with air masses that passed through the sectors 90$^{o}$ to 247.5$^{o}$ with an average concentration of 1.51 ng m$^{-3}$ and a maximum value of 1.71 ng m$^{-3}$ in the sector 135$^{o}$ – 157.5$^{o}$. The elevated GEM values within this sector are presumably mainly due to mercury sources in Poland, Romania, Bulgaria, Greece and from some Balkan countries. Coal combustion for production of electricity and domestic heating, but also smelters and other industrial activities are the likely sources of mercury. Based on these findings the sectors 0$^{o}$ to 90$^{o}$ and 247.5$^{0}$ to 360$^{0}$ were classified as mainly associated with background air that is not affected by regional mercury emission sources. This assumption is further supported by the PBM measurements as shown in Figure 3c. Particulate mercury has been proven to be a sensitive marker for polluted air containing mercury (Wängberg et al. 2003). The highest PBM concentrations were found in air from the south east sector as also is the case with GEM, but the difference in PBM concentration between different sectors is much greater. The GOM values show a similar pattern with elevated concentrations in the air from south east as shown in Figure 3d. However, in contrast to PBM relatively high GOM concentrations were also observed in air masses originating from north and east, indicating an additional source other than direct anthropogenic.

According to the classification of background and polluted air masses given above it was found that the Råö site receives air of background origin 59 % of the time with average concentrations of GEM, GOM and PBM equal to 1.36 ng m$^{-3}$, 0.76 pg m$^{-3}$ and 2.15 ng m$^{-3}$, respectively. Average GEM, GOM and PBM concentrations associated with polluted air masses yielded concentrations of 1.51 ng m$^{-3}$, 0.85 pg m$^{-3}$ and 5.71 ng m$^{-3}$, respectively.

### 3.3 Background and polluted air masses

The GEM, PBM and GOM values were divided into monthly averages and into background and polluted air masses according to the classification made in 3.2. The result is shown in Figure 4. Figure 4a show average monthly GEM concentrations. A seasonal variation with high values during the cold season is clearly depicted as well as a difference between air masses associated with background and polluted air. The higher values during winter are an effect of higher

mercury emissions from coal combustion etc. in combination with lower mixing heights during winter. Hence, it seems like both background and polluted air masses are influenced by anthropogenic emissions. A similar result is obtained with PBM as shown in Figure 4b, although the difference between background and polluted air masses is much more pronounced. The GOM data in Figure 4c, on the other hand, shows a quite different pattern with high values mostly occurring during the

warm season as shown in Figure 4c. Another difference is that the GOM concentration in background air may be higher than in polluted air, as is shown by the measurements obtained for May June and August. As is indicated by Figure 3d there is a coupling between elevated GOM concentrations and import of polluted air masses to the Råö site. However, the present results also strongly suggest an additional source, which is likely to stem from formation of GOM in the atmosphere. At the Råö site, elevated concentrations of GOM were only observed during daytime at dry and sunny weather with the highest

concentrations occurring during spring and summer. Diurnal patterns with peak concentrations in midday or early afternoon were observed and often repeated for several days during periods with fair weather. Like for ozone the diurnal variation is likely to be caused by nocturnal inversion at night, a phenomenon that occurs during clear sky conditions. GOM is then depleted due to deposition on the sea surface, vegetation and on wet aerosols. The inversion prevents GOM from above to mix with the air below until the next morning when the inversion is broken by the sun and GOM from above is transferred to

the ground through vertical mixing. The diurnal variation of ozone is well documented in the literature (Garland and Derwent, 1979; Zhang and Rao, 1999; Coyle et al., 2002). Råö is a coastal site which means that nocturnal inversions there are normally not as strong as at low altitude inland locations. Nevertheless, when comparing the daily GOM variation at the Rao site, with that of ozone one see that it almost perfectly coincides with the morning increase and evening decrease of ozone. Ozone is secondary pollutant that is formed in the atmosphere via atmospheric photolysis reactions and transported to

the Råö site rather than being formed locally. Likewise, GOM may also be formed in the troposphere by oxidation of GEM. The GOM formation is likely to be caused by a bromine-driven photolytic oxidation process (Donohoue et al., 2006; Obrist et al., 2011). GOM is probably formed at a slow rate in the atmosphere and due to its solubility it is efficiently being scavenged by atmospheric cloud droplets and therefore has a short residence time in the atmosphere. However, elevated concentrations may build up during dry atmospheric conditions in the free troposphere and transported to the lower

atmosphere by vertical mixing (Wängberg et al., 2007). From measurements at the Storm Peak Laboratory (in Rocky Mountains, Colorado, USA) at an elevation of 3200 m a.s.l., Fain et al., 2009 found evidence for formation and enrichment of GEM in the free troposphere as also has been observed in airborne measurements in the troposphere (Lyman and Jaffe 2012; Brooks et al., 2014). Elevated levels of GOM were observed both during day and night. GOM did not correlate with traces such as CO and aerosols, indicating that it was not primarily due to anthropogenic pollution sources. Relative

humidity was the dominant factor affecting the concentration with high GOM levels always present whenever relative humidity was below 40 to 50%. Fain et al., 2009 also found evidence for that GOM in the free troposphere may, due to subsidence, entrainment processes and deep vertical mixing, be transported throughout the troposphere and even into the boundary layer under dry cloudless conditions. That GOM, formed aloft in the atmosphere, may be transported to the ground is also supported by ground based measurements at three sites in Nevada, USA (Weiss-Penzias et al. 2009). The authors

found that in addition to in situ photochemical production, convective mixing and entrainment of dry air from the free troposphere during the day was an important contributor to the observed diurnal GOM pattern measured at especially one of Nevada sites, a remotely located area far away from urban sources of GOM. Like at Råö, the GOM concentrations in Nevada displayed a diurnal pattern that was positively correlated to ozone. The tropospheric origin of GOM measured at the Råö site is further supported by that the highest GOM values were observed in conjunction to import of air masses from the north that are not associated with major anthropogenic mercury sources. According to the trajectory analysis air masses with high GOM concentrations could be traced to the Norwegian Sea, the Greenland Sea and in some cases to the Barents Sea, i.e. sea areas north of Scandinavia.

**4 Conclusions**

Due to the remotely location of the Råö measurement station it receives background air most of the time. However, elevated concentrations with air masses coming from especially the south-east are noticeable. About one third of the air sampled at Råö has passed UK and Denmark. Going back some decades earlier a lot of coal was used in the UK for power production and domestic heating. As a result a lot of sulphur, and probably also mercury, was received and deposited in Scandinavia. The present investigation shows that little if any mercury origins from UK or Denmark today as a consequence of shift from coal to cleaner power production techniques. Mercury emissions from other countries, including Sweden, have also dramatically been reduced since 1990. During the period 1985 to 1989 the average TGM concentration at the Råö site was as high as 3.2 ng m$^{-3}$ (Iverfeldt et al., 1995).

At Råö, the airborne mercury species investigated all have a direct anthropogenic component, i.e. can be attributed to regional mercury sources, as well as of a background atmospheric origin. This is especially true regarding GEM, due to its long atmospherically residence time. The background concentration of GEM in the northern marine hemisphere is likely to resemble that measured at Mace Head. Particulate bound mercury has a much shorter atmospheric residence time, but small particles with an aerodynamic diameter of less than 2.5 μm may stay in the atmosphere long enough to contribute to a background level of PBM. Due to the normally short atmospheric life time of GOM one may not expect to receive air masses of remote origin with substantial concentrations of GOM by the same means as with GEM and PBM. According to the kinetics of the atmospheric oxidation process of GEM to GOM (Donohoue et al., 2006), elevated concentrations of GOM are not likely to be formed locally in the air near background measurement sites. Hence, elevated GOM concentrations measured in background air at the Råö site presumably originate from air masses enriched in mercury from the free troposphere, something that occurs during dry conditions. As far as we know, this is for the first time real evidence of tropospheric formed GOM has been observed at ground based measurements in Europe.

## 5 Acknowledgment

The financial support from The European Commission, FP7 (contract no. 26511) is gratefully acknowledged.

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

**Tables**

**Table 1.** Median and average mercury concentrations measured at some Northern European sites

| Measurement sites | GEM/TGM | GOM | PBM | Period | Reference |
|---|---|---|---|---|---|
| | median | median | median | | |
| | ng m$^{-3}$ | pg m$^{-3}$ | pg m$^{-3}$ | | |
| Råö | 1.41 | 0.23 | 2.21 | 2012 - 2015 | This work |
| Mace Head* | 1.48 | - | - | 2012 - 2015 | |
| Waldhof | 1.61 | 1.0 | 6.30 | 2009 - 2011 | Weigelt et al., 2013 |
| | Average | Average | Average | | |
| | ng m$^{-3}$ | pg m$^{-3}$ | pg m$^{-3}$ | | |
| Råö | 1.42 ± 0.20 | 0.80 ± 1.6 | 3.6 ± 4.5 | 2012 - 2015 | This work |
| Mace Head* | 1.48 ± 0.14 | - | - | 2012 - 2015 | |
| Aucgencort Moss | 1.40 ± 0.19 | 0.43 ± 1.7 | 3.1 ± 5.3 | 2009 - 2011 | Kentisbeer, 2014 |
| Harwell* | 1.45 ± 0.24 | - | - | 2012 - 2013 | Kentisbeer, 2015 |

*Total gaseous mercury measured with Tekran instruments, calculated for the same periods as in this work.

a)

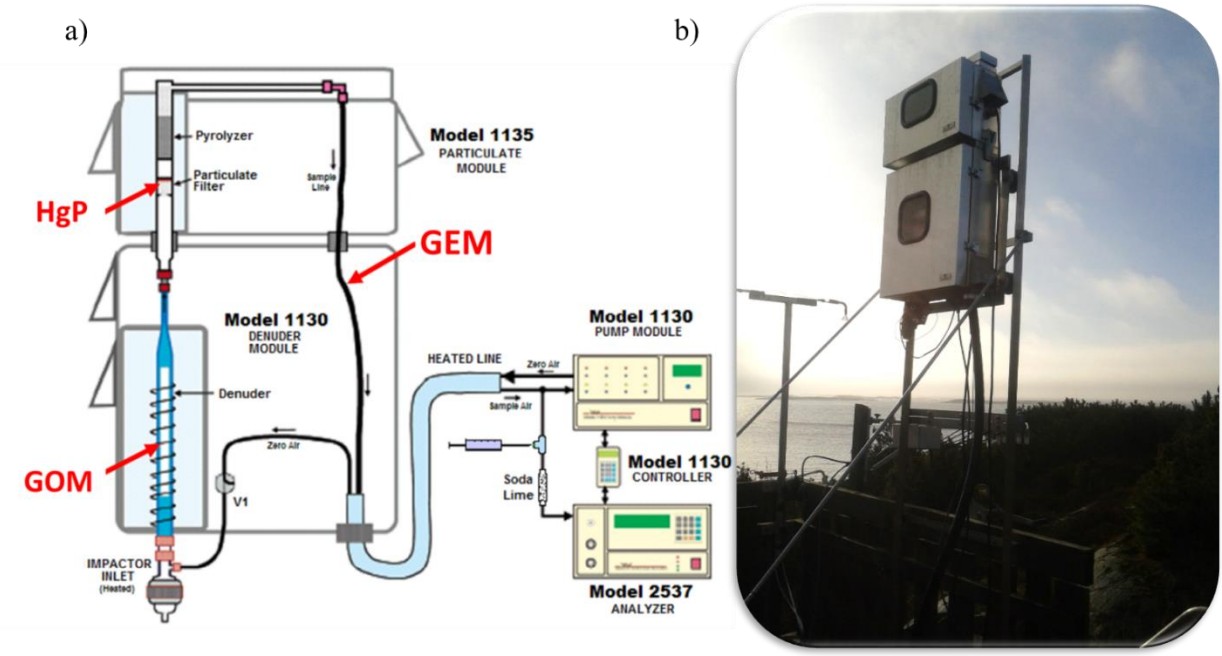

b)

**Figure 1**. a) a schematic overview of the Tekran mercury speciation system 1130/35 and the 2537 mercury analyser, b) picture of the instrument operating at the Råö station.

a)

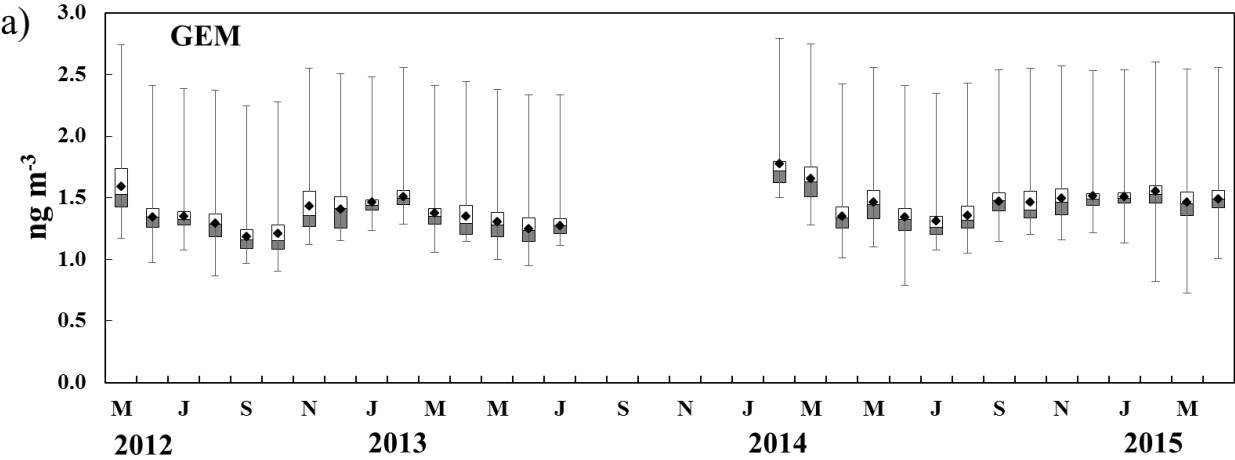

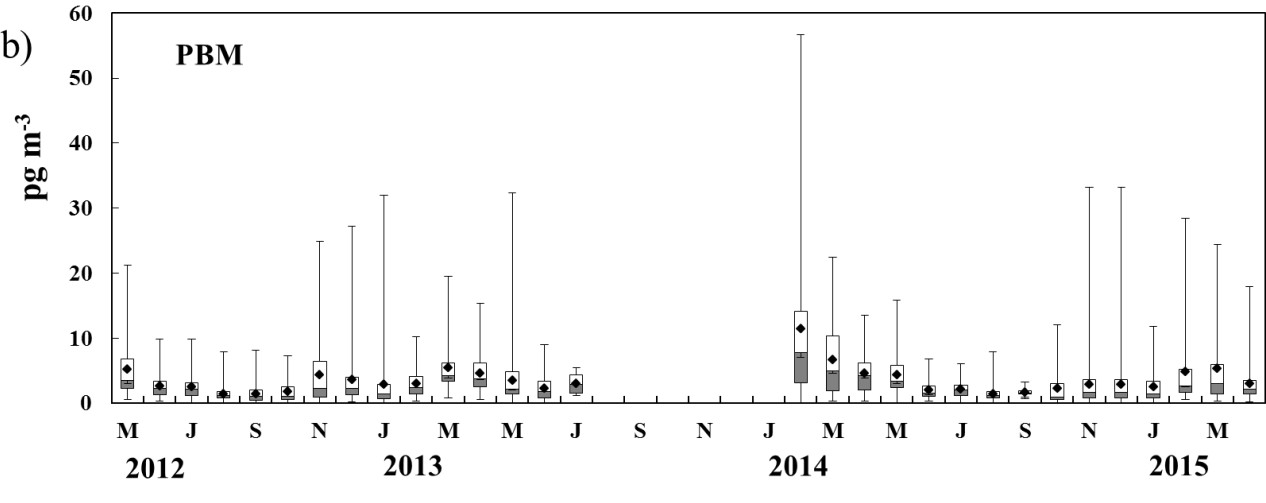

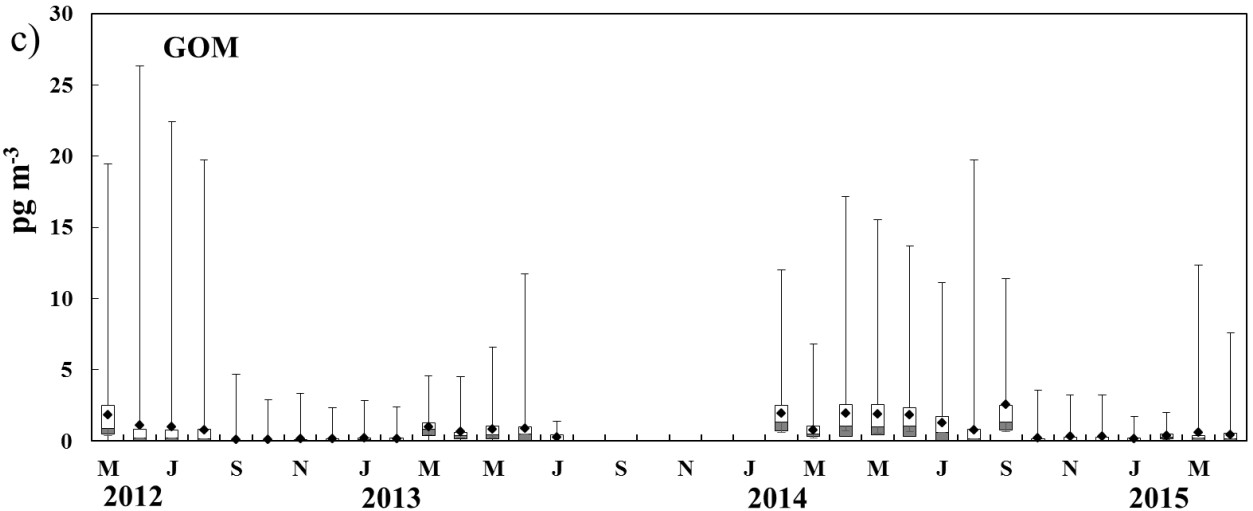

**Figure 2.** Boxplots of measured a) GEM, b) PBM and c) GOM at Råö/Rörvik from May 2012 to July 2013 and from February 2014 to April 2015. Diamonds show average values. Upper and lower boxes show 75 % and 25 % percentiles, respectively with median values in between. Upper and lower whisker bars shows maximum and minimum values, respectively.

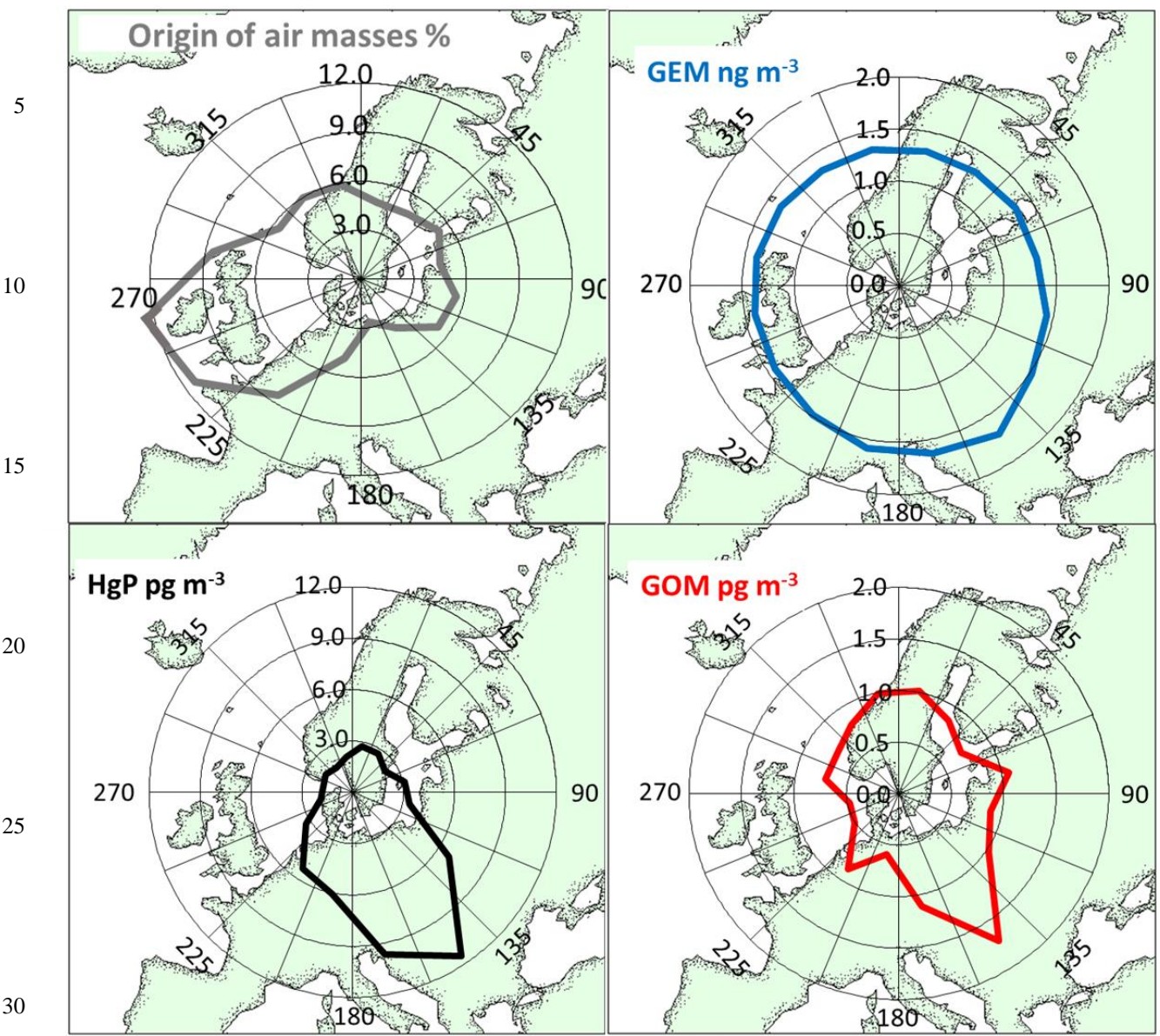

**Figure 3.** Results of the back trajectory analysis. a) origin of air masses reaching the Råö site presented as relative frequency (%) within each sector. b), c) and d) shows GEM, PBM and GOM concentrations, respectively as function of origin of air masses entering the measurement site.

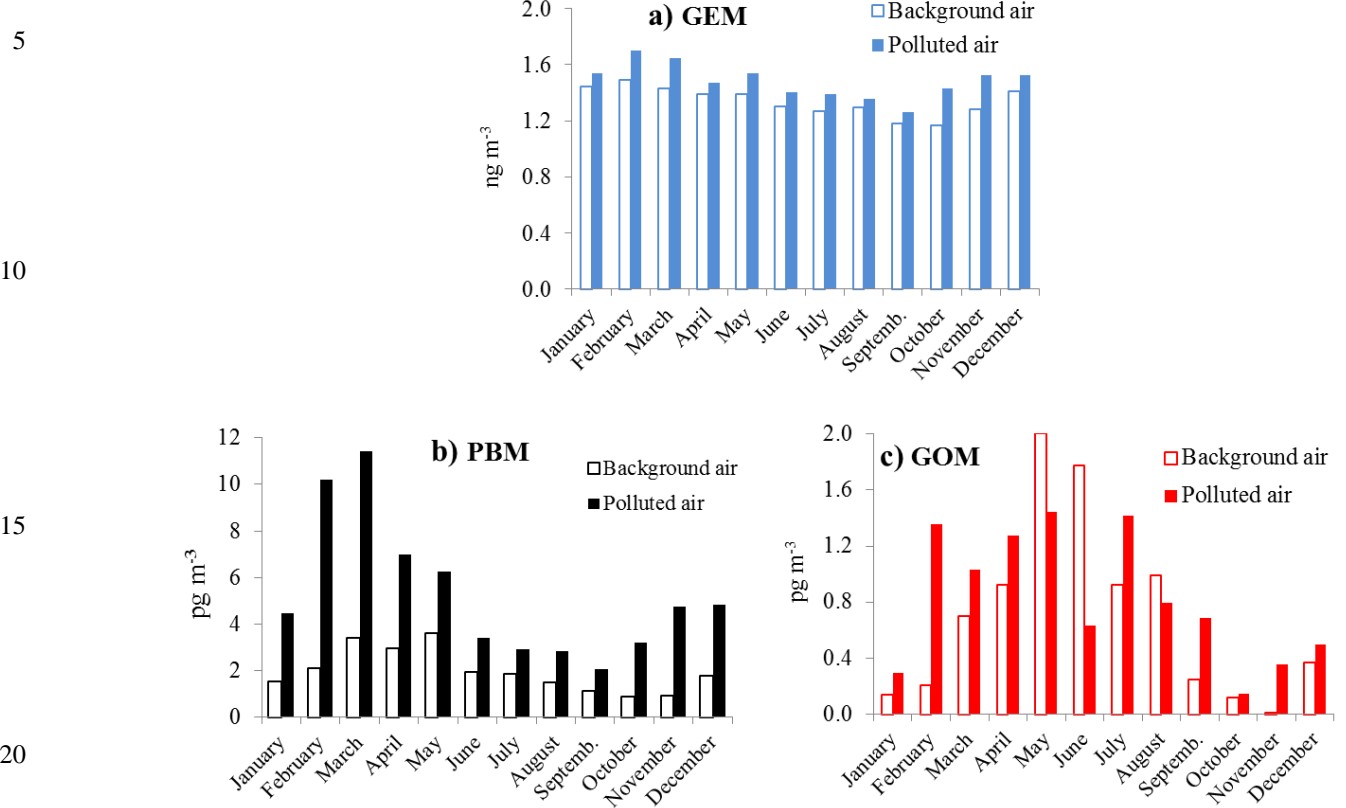

**Figure 4.** Average mercury air concentrations as function of month and origin of air masses at the Råö measurement site. The values represent monthly averages of all data from the measurements made during 2012-05-15 to 2015-04-29.