# Peer review of "Airborne mercury species at the Råö background monitoring site in Sweden: Distribution of mercury as an effect of long range transport"

_Atmospheric Chemistry and Physics, 2016_

## Referee Comment (RC1) · Anonymous Referee #1 · 5 Jul 2016

Review of acp-2016-526 "Airborne mercury species at the Råö background monitoring site in Sweden: Distribution of mercury as an effect of long range transport"

Overview: The submitted manuscript deals with the measurement and analysis of speciated mercury measurements, performed in two periods between 2012 and 2015 at Råö which is a background site in southern Sweden. For the speciation measurements a Tekran 1130/1135 unit and a Tekran 2537 mercury analyzer was used. The measurements were carried out within the Global Mercury Observation System (GMOS) Project. The analysis of the measurement data focus on the comparison to other European measurements, seasonality, and air mass origin. Within different national and international measurement programs (e.g. AMNet, CAMNet, GMOS,...) many mercury

measurements were made all over the world using equal measurement technique and similar data analysis. Therefore the innovation, new/unique technique or new findings are missing somehow. Nevertheless, the presented dataset is important and should be considered for publication, because it represents the second longest mercury speciation dataset in Europe. This means the dataset will be of high interest for future mercury model studies to validating regional and global atmospheric chemistry transport models. However, there are some substantial limitations of the dataset and some clarification is needed before considering for publication:

General comment:

On Page 3 Line 30 and Page 4 line 11 and 12 the lower detection limits are given to be 0.014 ng m-3 for GEM (sampling time of 5 min with 1 l/min), 0.11 pg m-3 for PBM and 0.23 pg m-3 for GOM (both with sampling time of 180 min with 10 l/min). According to Swatzendruber et al., 2009 (Atmos. Environ., 43, 3648–3651) and Slemr et al., 2016 (ACP, doi: 10.5194/amt-9-1-2016) the Tekran analyzers have significant problems with the internal raw data dump peak integration when the total amount of mercury on the gold traps is below 2 pg. In fact the Tekran internal peak integration underestimates the measured concentration by about 20% when the mercury load is 2 pg and by > 40% when the mercury load is 1 pg (exponential increase; see Fig. 3 in Slemr et al., 2016). The detection limits given in the reviewed manuscript represent mercury loads of 0.07 pg for GEM, 0.2 pg for PBM and 0.4 pg for GOM. Assuming a minimum lower mercury load of 2 pg and the flow rates and sampling times given in the manuscript would result in lower detection limits of 0.4 ng m-3 for GEM and 1.1 pg m-3 for PBM and GOM (all still with an uncertainty of 20%). Considering this, the GOM mean given in Table 1 would be below the detection limit. The good news is that if the raw data dump is available, this error can be corrected/avoided. Did the authors record the Tekran 2537 raw data dump and checked for correct peak integration of the Tekran internal integration algorithm in case of low concentration? If so, did they reanalyzed the data using an external integration algorithm? If not, would it be possible to check the unit

for underestimation of the mercury concentration as a function of mercury load on the traps and to correct the data (like in Slemr et al., 2016)?

Specific comments:

P1 L8 to P2 L2: The statements in the introduction should be underlined with some literature concerning properties of Hg, atmospheric specification, lifetime, deposition, transformation, environmental- and health effects.

P2 L4-6: According to Pirrone et al. 2010 (doi:10.5194/acp-10-5951-2010) and Song et al., 2015 (doi:10.5194/acp-15-7103-2015) coal combustion is the biggest anthropogenic Hg source. It is recommended to change order to importance of source and cite the above mentioned papers, too.

P2 L31: What is the definition of a "real" background site? Dose this imply that the other measurement sites in GMOS, AMNet, CAMNet are no real background sites? This statement is in contrast to a statement in the conclusions (P7 L24).

P2 L31 to P3 L1: Even more important than the wind speed is the main wind direction.

P3 L19-30: The Tekran 1130/1135 setup was already described in many publications and the instrument manual. So this description can be shortened by mention the measurement units, give the setup for the temporal resolution and for the interested reader link to the manual and/or further studies.

P4 L10: Not only the lower detection limit, but also an estimation to the measurement accuracy of GEM, GOM and PBM should be given. As the installation and analysis algorithm is probably similar to those described in Weigelt et al. 2013, the authors could adopt this estimation. However, Gustin et al., 2013 (Do we understand what the mercury speciation instruments are actually measuring? Results of RAMIX, Environ. Sci. Technol., 47, 7295–7306) and Gustin et al., 2015 (Measuring and modeling mercury in the atmosphere: a critical review, Atmos. Chem. Phys., 15, 5697–5713, doi:10.5194/acp-15-5697-2015) should be considered, too.

P4 L14-20: As trajectory calculations are used in several papers and are a quite common way to check of air mass origin, in my opinion there is no need to motivate and introduce trajectory calculations in general. Just as a suggestion, this section could start with something like "Three day back trajectories were calculated to derive information on transport path and potential sources of the air mases measured at Råö. For the calculation the Hybrid Single-Particle...continue line 21" If the time information (three day backward) is shift to the beginning of the section, the sentence on L 20-31 can be skipped.

P5 L1: This section should contain a short statement on the uncertainty of the trajectories, too. For example the authors could give a rough estimation what was the average difference between the trajectories starting at 10, 50, 100, 250, and 500 m (e.g. what was the horizontal difference in the trajectory end points).

P5 L16: It is mentioned Mace Head is a GMOS site, too and according to the GMOS web page the measurements are ongoing. So the authors could also compare the same period.

P5 L16-21: The 1 $\sigma$ standard deviation given in Tab. 1 is much higher than the difference in the average data. The measurement uncertainty should be somewhat smaller than 1 $\sigma$ (I guess something between 10 and 20%), but even higher than the difference. Considering the measurement uncertainty, Råö, Mace Head, Aucgencort Moss, and Harwell are equal.

P5 L22-24: Please consider the general comment to the lower detection limit. According to Fig. 3 in Slemr et al., 2015, GOM might be underestimated by more than 30% (average) and 60% (median). If the low GOM concentrations at Råö are real, the authors could additionally discuss why the average GOM is four times the median but for PBM the average is only 60% higher. Furthermore it is interesting the GOM/PBM ratio is more or less similar at Råö, Aucgencort Moss, and Waldhof.

P6 L30-32: Not necessarily influenced by anthropogenic emissions, could be also nat

ural emissions. Or did the authors measure additional tracers, indicating an anthropogenic origin (e.g. SF6)? The authors should consider also a seasonal changing planetary boundary layer (PBL) height. During summer the height of the PBL and therefore the mixing volume is higher than during summer.

P7 L1: According to the above given general comment during winter GOM might be underestimated.

P7 L6: "proposed earlier" → Where? In 3.2 the authors just mention other regions but not the free troposphere

P7 L9: cloud droplets and aerosol particles

P7 L12-14: Did the trajectory analysis confirmed that air masses with elevated GOM came from the free troposphere, whereas air masses with low GOM traveled inside the PBL all the time? In principle during summer GOM could be formed also in the local PBL around the measurement station. The authors can check this by plotting a daily cycle for the summer month. A three hour sampling interval and six samples a day should roughly resolve a daily cycle when averaging all summer data. If a maximum at noon/early afternoon is visible, it is probably local production because transport from the free troposphere would take several hours and therefore no maximum would be expected at noon/early afternoon.

P7 L24: If all investigated mercury species have direct anthropogenic component, in contrary to the statement on P2 L31 Råö is not a real background station.

P7 L25: "…regarding GEM,…" Looking on Fig. 3b,c and Fig 4a,b the difference is even more pronounced for PBM. The increase is around 10 to 20% for GEM, but 100 to 200% for PBM. So the sentence should be: "This is especially true for PBM,…"

P7 L28: The authors should be more precise what did they mean with small particles (e.g. smaller 10, 100, 1000 nm; or nucleation mode, Aitken mode, accumulation mode coarse mode particles).

P7 L30 to P8 L1: As indicated in a previous comment, it is recommended to test this assumption by analyzing the GOM measurements for the presence of a daily cycle in summer. Are there any tracers available, indicating the air was coming from the free troposphere (low rH, low aerosol concentration,...)? If not, at least a statement should be made that the trajectory analysis showed that high GOM concentrations were linked to free tropospheric air origin (was not stated before).

P8 L1-2: The meaning of this sentence is not clear. Do the authors mean that these are the first measurements in Northern Europe giving evidence that GOM is formed in the free troposphere? Did ground based and airborne measurements in other regions observed a similar behavior of GOM (e. g. at other (high elevated) GMOS sites, or studies on vertical distribution of GOM by Lyman and Jaffe 2012, Nature Geosci., 5, 114–117, 2012, and Brooks et al., 2014; Atmosphere, 5, 557–574)?

Table 1: On page 5 Line 30 it is written that 50% of the GOM measurements were below the detection limit. How did the authors consider measurements below the detection limit for the median/average calculation? Please explain in the text. For Waldhof the authors give only the median and for Aucgencort Moss and Harwell only the average concentration. Would it be possible to calculate median and average for all stations?

Technical comments:

P1 L8: "to the" instead of "tot the"

P1 L9: "were" instead of "was"

P2 L13-14: Framework Program (FP7)

P3 L5: were instead of was

P3 L6: please define acronym CVAFS

P3 L12: upstream instead of up streams

P4 L31: "Six such back trajectories were calculated..." maybe better "Six back trajectory ensembles were calculated..."

P5 L4: acronym VBL needs to be defined

P6 L9: Fig. 3a should be Fig 3b

P6 L17: maybe better write "in air from the south east sector" instead of "in the south east sector"

P6 L18: maybe better write "in the air from south east" instead of "in the south east"

P12 Figure 1a: I'm not sure it is necessary to show the 1130/1135 flow chart, as it is a standard instrument and the flow chart is given in the manual. If the authors want to show the flow chart, at least a link to the source of the graph is needed. HgP should be PBM, because in the text only PBM is used.

P12 Figure 1b: To have a more professional look, the 3d effect with round reflecting edges should be avoided. More space between a) and b) is needed.

P14 Fig 4: It is recommended to avoid the color changeover in the individual columns. Instead two different colors should be used to differ between background air and polluted air. The labeling at the x-axis in a), b), and c) is not correct: Januari and Februari should be January and February. If the y-scale in Fig. a) starts at 1 ng m-3, the seasonality would be better to see.

---

## Referee Comment (RC2) · J. Knœry (Referee) · 25 Aug 2016

**Review of acp-2016-526 "Airborne mercury species at the Råö background monitoring site in Sweden: Distribution of mercury as an effect of long range transport » by I. Wängberg, M. G. Nerentorp Mastromonaco, J. Munthe, and K. Gårdfeldt**

**Excellent paper. Could be published with minor corrections suggested below.**

**A: -requests for improvements:**

Draw a figure (simple box model) to illustrate your argument that the free tropospheric GOM is a likely source of "excess" GOM observed at Rao. (line 10-15, p7)

Attempt to compare with the La Seyne sur Mer data presented in Marusczak et al. (2014)

**B: - suggestions for improvement**

Title :

Airborne mercury species at the Swedish Rao monitoring site : their distributions are affected by long-range transport.

Abstract :

Within the EU-funded GMOS project, …

mid-May

line 11 : remote location

south-east

background, free tropospheric air

Introduction

Line 26 : particulate-bound mercury

Page2

Line1 : chain, which occurs frequently in marine and freshwater ecosystems

Line 10: bedrock, and their contributions to… are estimated…

Line 15: Tekran speciation unit was used…

Line 24: the detail and amount of comparison given in the text does not warrant the amout of "teasing" performed in the introduction.

Line 30 GMOS master

Line 31: considered a real background site: please provide references or additional information.

Line 1: give percentiles/extrema/ standard deviations associated with average meteorological parameters.

Line 15: were obtained

Line 17: every four hours, three-hour average PBM and GOM values are btained, together….

Line 27: to ensure that all oxidized

Line 30: quantified by the …

Page4

Line 1: laboratory-built

Line 6: Once the blanks are at the appropriate level, PBM and GOM were always detected…

Line 11: do reference the GMOS SOP or link to the web-site.

Line 5: could be associated with each.

Line 24: probably because the Waldhof

Line 30 : or GOM. Close to…

Line4: simplify sentence

Line 11: mercury sources in Poland, Romania, ..

Line 12: electricity and domestic heating, but also

Line 25: and 5.71 ng m^-3

Line 31: influenced by …

Line 7: bromine-driven photolytic oxidation (Donohoue et al., 2006)

Line 8: formed at a slow rate

Line 17: south-east… the air sampled at Rao has …

Line 19: domestic heating

Line 24 At Rao, the airborne…

Line 29: normally short atmospheric… of GOM, one …

Line 31, GOM are not likely…

Page 11:

Table 1: do separate better the upper table (Median) and lower part (Average)

Fig 4: use segmented line plots rather than bars. It would allow to plot means and medians on the same figure. If bars MUST be used, avoid gradient fills. January and February are misspelled.

---

## Author Comment (AC1) · 4 Oct 2016

Answer to Anonymous Referee #1

Airborne mercury species at the Råö background monitoring site in Sweden: Distribution of mercury as an effect of long range transport Author(s): I. Wängberg et al. MS No.: acp-2016-526 MS Type: Research article Special Issue: Global Mercury Observation System – Atmosphere (GMOS-A)

The authors thanks for the knowledgeably and thorough work made by the Anonymous Referee #1. The manuscript has been very much improved after considering the comments from the Referee. Best regards, Ingvar Wängberg

[Figure]

Comments from referee #1

Overview: The submitted manuscript deals with the measurement and analysis of speciated mercury measurements, performed in two periods between 2012 and 2015 at Råö which is a background site in southern Sweden. For the speciation measurements a Tekran 1130/1135 unit and a Tekran 2537 mercury analyzer was used. The measurements were carried out within the Global Mercury Observation System (GMOS) Project. The analysis of the measurement data focus on the comparison to other European measurements, seasonality, and air mass origin. Within different national and international measurement programs (e.g. AMNet, CAMNet, GMOS,. . .) many mercury measurements were made all over the world using equal measurement technique and similar data analysis. Therefore the innovation, new/unique technique or new findings are missing somehow. Nevertheless, the presented dataset is important and should be considered for publication, because it represents the second longest mercury speciation dataset in Europe. This means the dataset will be of high interest for future mercury model studies to validating regional and global atmospheric chemistry transport models. However, there are some substantial limitations of the dataset and some clarification is needed before considering for publication:

General comment: On Page 3 Line 30 and Page 4 line 11 and 12 the lower detection limits are given to be 0.014 ng m-3 for GEM (sampling time of 5 min with 1 l/min), 0.11 pg m-3 for PBM and 0.23 pg m-3 for GOM (both with sampling time of 180 min with 10 l/min). According to Swatzendruber et al., 2009 (Atmos. Environ., 43, 3648–3651) and Slemr et al., 2016 (ACP, doi: 10.5194/amt-9-1-2016) the Tekran analyzers have significant problems with the internal raw data dump peak integration when the total amount of mercury on the gold traps is below 2 pg. In fact the Tekran internal peak integration underestimates the measured concentration by about 20% when the mercury load is 2 pg and by >40% when the mercury load is 1 pg (exponential increase; see Fig. 3 in Slemr et al., 2016). The detection limits given in the reviewed manuscript represent mercury loads of 0.07 pg for GEM, 0.2 pg for PBM and 0.4 pg for GOM.

Assuming a minimum lower mercury load of 2 pg and the flow rates and sampling times given in the manuscript would result in lower detection limits of 0.4 ng m-3 for GEM and 1.1 pg m-3 for PBM and GOM (all still with an uncertainty of 20%). Considering this, the GOM mean given in Table 1 would be below the detection limit. The good news is that if the raw data dump is available, this error can be corrected/avoided. Did the authors record the Tekran 2537 raw data dump and checked for correct peak integration of the Tekran internal integration algorithm in case of low concentration? If so, did they reanalyzed the data using an external integration algorithm? If not, would it be possible to check the unit C2 for underestimation of the mercury concentration as a function of mercury load on the traps and to correct the data (like in Slemr et al., 2016)? Answer: The GMOS community are aware of this integration malfunction regarding airborne measurements. However the author of the present paper did not fully realise to what extent it may affect ground site GOM and PBM measurements. Hence, the Standard Operational Procedure (SOP) developed for GMOS, which harmonises with the SOPs used in the USA and Canada was used at all sites. Of course it would have been interesting to see how great this integration bias influences the present measurements. But, we have no "raw data dump" values other than from tests made in connection to maintenance. Unfortunately, such tests were never made with low concentrations. As also mentioned by the referee the integration problem at low concentrations is not the only issue regarding the Tekran measurements. The GOM measurements are also expected to be erroneous due to interference with ozone for example. I think we must recognize the operational nature of the Tekran GOM and TPM measurements, i.e. the result is method dependent. However, the present work focuses on the origin of air masses transporting GEM, PBM and GOM to the Råö measurement site and most of the conclusions made are not dependent on the actual level of GOM and PBM. The present result, presented in Table 3 compares well with other measurements made in Northern Europe, using the same instrumental measurement technique and gives a hint on mercury concentrations in this region although the levels of GOM and PBM probably are underestimated. Comments on the difficulties with low concentrations will

be commented in the Method chapter.

Specific comments: P1 L8 to P2 L2: The statements in the introduction should be underlined with some literature concerning properties of Hg, atmospheric specification, lifetime, deposition, transformation, environmental- and health effects. Answer: Most of this is now included.

P2 L4-6: According to Pirrone et al. 2010 (doi:10.5194/acp-10-5951-2010) and Song et al., 2015 (doi:10.5194/acp-15-7103-2015) coal combustion is the biggest anthropogenic Hg source. It is recommended to change order to importance of source and cite the above mentioned papers, too. Answer: It seems like Song et al. 2015 do not at all mention specific anthropogenic mercury sources. The mercury emission data presented by Pirrone et al. 2010 stems from earlier estimates (Pacyna et al., 2006 and Streets et al., 2009). Today it exist evidence for that artisanal and small scale gold mining constitutes the largest anthropogenic mercury source. Pacyna et al., 2016 estimated the Hg emission from artisanal gold mining based on consumption patterns and assumptions on the emissions to air from different mining methods employed in different regions. Although these estimates are uncertain the authors suggest that during year 2013 as much as 37 % of the direct anthropogenic emission of mercury was due to artisanal gold mining which can be compared to 25 %, which according to the authors, constituted the global contribution from combustion of coal.

P2 L31: What is the definition of a "real" background site? Dose this imply that the other measurement sites in GMOS, AMNet, CAMNet are no real background sites? This statement is in contrast to a statement in the conclusions (P7 L24). Answer: A background site is a site where the mercury concentration measured is representative for an as large area around the measurement site as possible. Regarding the Råö station this includes a large part of the southern Nordic region. Within the GMOS project the aim is to choose measurement places representative for as large areas as possible and I assume that this also the case within the other background monitoring programs mentioned in the present comment. Hence, with this definition almost all

measurement sites could be considered to be real background sites as long as they are not located in or close to urban areas. But, their importance and usefulness concerning evaluation of the global impact on anthropogenic mercury emissions etc. varies of course with how large areas the data gathered might represent. Therefore, in the present work, a lot of effort was put on to distinguish between air masses reaching the Råö measurement site coming from European mercury emission areas and those originating from areas not directly associated with polluted air. This evaluation showed that the Råö site received air of background origin 59 % of the time and hence, is a real background site in that respect.

Is this statement in contrast to a statement in the conclusions (P7 L24)? Answer: No it is not, but the sentence on P7 L24 need to be rewritten to be made clearer. As is shown in the manuscript, Figure 4, regarding GEM, both air associated with polluted sectors as those linked to clean sectors show a seasonal pattern with slightly higher values during the cold season in comparison to summer. Obviously this reflect the higher emissions during winter in combination with lower mixing heights when the combustion of coal is high, but as shown in Figure 4 this pattern is seen in both air from polluted sectors and clean sectors. The same is also true regarding PBM as shown in Figure 4.

P2 L31 to P3 L1: Even more important than the wind speed is the main wind direction. Answer: The predominantly wind direction to the Råö site is south west as indicated in Figure 3a. This information is now also included in the text.

P3 L19-30: The Tekran 1130/1135 setup was already described in many publications and the instrument manual. So this description can be shortened by mention the measurement units, give the setup for the temporal resolution and for the interested reader link to the manual and/or further studies. Answer: The text will be shortened in this respect according to the referee's comment. However, the manual is not free to download.

P4 L10: Not only the lower detection limit, but also an estimation to the measurement

accuracy of GEM, GOM and PBM should be given. As the installation and analysis algorithm is probably similar to those described in Weigelt et al. 2013, the authors could adopt this estimation. However, Gustin et al., 2013 (Do we understand what the mercury speciation instruments are actually measuring? Results of RAMIX, Environ. Sci. Technol., 47, 7295–7306) and Gustin et al., 2015 (Measuring and modeling mercury in the atmosphere: a critical review, Atmos. Chem. Phys., 15, 5697–5713, doi:10.5194/acp-15-5697-2015) should be considered, too. Answer: In lack of sufficient reference materials, determining the Tekran measurement accuracy is difficult as also is mentioned by Weigelt et al. 2013. They were not able to present the correctness of their measurements, but did estimate the reproducibility of the measurements based on running two Tekran instruments in parallel. The resulting reproducibility was $\pm$ 12 %. This result could also be applicable in the present case as suggested by the Referee. The operational nature of the "mercury speciation" measurements is to some length mentioned in introduction, but will further be emphasised, mentioning the results of Gustin et al.

P4 L14-20: As trajectory calculations are used in several papers and are a quite common way to check of air mass origin, in my opinion there is no need to motivate and introduce trajectory calculations in general. Just as a suggestion, this section could start with something like "Three day back trajectories were calculated to derive information on transport path and potential sources of the air mases measured at Råö. For the calculation the Hybrid Single-Particle. . .continue line 21" If the time information (three day backward) is shift to the beginning of the section, the sentence on L 20-31 can be skipped. Answer: This section has been shortened.

P5 L1: This section should contain a short statement on the uncertainty of the trajectories, too. For example the authors could give a rough estimation what was the average difference between the trajectories starting at 10, 50, 100, 250, and 500 m (e.g. what was the horizontal difference in the trajectory end points). Answer: Since the purpose of the trajectory evaluation was to identify the origin of air masses reaching the Råö site in terms of the association with different wind sectors as shown in Figure 3. The results from back trajectories with starting height 10, 100, 250 and 500 m were compared during the first measurement period May 2012 to July 2013. The wind sector plots obtained are not exactly the same in all details, but gave quantitatively the same information. Hence, the information gathered from the trajectory evaluations seem to a large degree to be independent on starting height. A detailed comparison of back trajectories with starting heights 100 and 250 m were performed using all data from both measurement periods, May 2012 to July 2013 and February 2014 to May 2015. The back trajectories with start heights 100 and 250 m gave quantitatively the same information regarding origin of air masses that reached the Råö site during the measurements.

P5 L16: It is mentioned Mace Head is a GMOS site, too and according to the GMOS web page the measurements are ongoing. So the authors could also compare the same period. Answer: The data presented from Mace Head is now from the same time periods as the present work.

P5 L16-21: The 1 ïĄş standard deviation given in Tab. 1 is much higher than the difference in the average data. The measurement uncertainty should be somewhat smaller than 1 ïĄş (I guess something between 10 and 20%), but even higher than the difference. Considering the measurement uncertainty, Råö, Mace Head, Aucgencort Moss, and Harwell are equal. Answer: I am not sure I fully understand this point. The 1 ïĄş standard deviation mostly reflects the temporal variation of TGM or GEM concentrations at the sites, i.e. as such not a measure of uncertainty.

P5 L22-24: Please consider the general comment to the lower detection limit. According to Fig. 3 in Slemr et al., 2015, GOM might be underestimated by more than 30% (average) and 60% (median). If the low GOM concentrations at Råö are real, the authors could additionally discuss why the average GOM is four times the median but for PBM the average is only 60% higher. Furthermore it is interesting the GOM/PBM ratio is more or less similar at Råö, Aucgencort Moss, and Waldhof. P6 L30-32: Not necessarily influenced by anthropogenic emissions, could be also nat-ural emissions. Or did the authors measure additional tracers, indicating an anthropogenic origin (e.g. SF6)? The authors should consider also a seasonal changing planetary boundary layer (PBL) height. During summer the height of the PBL and therefore the mixing volume is higher than during summer. Answer: The difference between average and median values of GOM in comparison to that of PBM is simply a cause of that the GOM data contain much more zero values than that of PBM. About 38 % of the GOM measurements were equal to 0.000 pg m-3 (uncorrected values from the Tekran instrument) whereas for PBM the corresponding figure only is 1.6 %. I think the difference between average and mean values is inevitable regarding GOM as the concentrations often are close to zero. This occurs when it rains for example, something that does not affect PBM to the same degree. In the case with GEM, median and average values are almost identical, see the values for Råö and Mace Head. Hence, the GEM concentrations are almost normal distributed, which makes sense considering the background character of these values.

P7 L1: According to the above given general comment during winter GOM might be underestimated. Answer: Yes, but this does not prevent us from making the general conclusion that the GOM concentration tend to be higher during the warm season than during winter.

P7 L6: "proposed earlier" ! Where? In 3.2 the authors just mention other regions but not the free troposphere Answer: As has been proposed earlier refers to the reference Wängberg et al., 2007, in the end of the chapter. The whole chapter is now rewritten and hopefully more informative and easier to read.

P7 L9: cloud droplets and aerosol particles Answer: The text has been improved in this chapter, see my answer on P7 L6, above.

P7 L12-14: Did the trajectory analysis confirmed that air masses with elevated GOM came from the free troposphere, whereas air masses with low GOM traveled inside the

PBL all the time? In principle during summer GOM could be formed also in the local PBL around the measurement station. The authors can check this by plotting a daily cycle for the summer month. A three hour sampling interval and six samples a day should roughly resolve a daily cycle when averaging all summer data. If a maximum at noon/early afternoon is visible, it is probably local production because transport from the free troposphere would take several hours and therefore no maximum would be expected at noon/early afternoon. Answer: Thanks to the comments from the referee, we realise that this part needs a more thorough presentation. In fact, we actually see a diurnal behaviour with peak concentrations of GOM in midday or early afternoon during spring, summer and early fall at the Råö site. Elevated concentrations of GOM are only observed during daytime at dry and sunny weather and the diurnal pattern may be repeated for several days during periods with fair weather conditions. This observation together with the diurnal variation of GOM may lead to the assumption that GOM is locally formed as a course of atmospheric photochemistry. However, we interpreted these observations in a different way. The GOM is likely to be transported to the site with air masses enriched in oxidised mercury. The source of GOM may be of direct anthropogenic origin as observed when the air comes from European mercury source areas or stem from air enriched in oxidised mercury that slowly was built up due to oxidation in the atmosphere when transported to the site. Whether the origin of the atmospheric formed GOM is the PBL or the free troposphere could not be concluded in this work. However, GOM formation in the free troposphere has been confirmed by other investigations, see the new comments in the text. The diurnal variation in turn is likely to be coursed by nocturnal inversion at night a phenomenon that occurs during clear sky conditions. GOM is then depleted due to deposition on the sea surface, vegetation and on wet aerosols. The inversion prevents GOM from above to mix with the air below until the next morning when the inversion is broken by the sun and GOM from the PBL above is transferred to the ground through vertical mixing. This phenomenon is well known and described in the literature and has been observed for ozone, a secondary air pollutant that are formed in the atmosphere as a course of atmospheric photolysis reactions (Garland and Derwent, 1979; Zhang and Rao, 1999; Coyle et al., 2002). Råö is a coastal site which means that nocturnal inversions there are normally not as strong as at low altitude inland locations. Nevertheless, when comparing the daily GOM variation at the Rao site, with that of ozone one see that it almost perfectly coincides with the morning increase and evening decrease of ozone. GOM like ozone are only to a limited extent likely to be formed locally in Southern Sweden. Most of the mercury and ozone measured at the Råö site is imported from elsewhere. This interpretation is strongly supported by the ozone observations and the coupling to weather condition as well as by the result shown in Figure 4 since the highest average monthly concentrations of GOM are observed in sectors not associated with polluted air. Hence, GOM is to a significant degree a secondary pollutant that is formed in the PBL or in the free troposphere and is transported to the Råö site.

P7 L24: If all investigated mercury species have direct anthropogenic component, in contrary to the statement on P2 L31 Råö is not a real background station. Answer: Råö is a background site in the sense it receives background air most of the time. The atmospheric content of mercury is due to natural as well as anthropogenic emissions, with the latter being the most dominant. Hence, most of the atmospheric mercury is of anthropogenic origin. However, it is possible to distinguish between mercury in air masses coming directly from mercury sources from that in background air. The point I wanted to make here is that the annual variation in mercury also is discernible in background air, although the higher concentrations in winter are probably not only due to higher mercury emissions but also to lower mixing heights as mentioned above. The text on this part is now rewritten to be more clearly.

P7 L25: "... regarding GEM,. . ." Looking on Fig. 3b,c and Fig 4a,b the difference is even more pronounced for PBM. The increase is around 10 to 20% for GEM, but 100 to 200% for PBM. So the sentence should be: "This is especially true for PBM,. . ." Answer: This statement is meant to refer to the evenly distribution of GEM in the so called clean sectors.

P7 L28: The authors should be more precise what did they mean with small particles (e.g. smaller 10, 100, 1000 nm; or nucleation mode, Aitken mode, accumulation mode coarse mode particles). Answer: The sentence is now rewritten in the following way: Particulate bound mercury has a much shorter atmospheric residence time, but small particles, less than 2.5 $\mu$m, may stay in the atmosphere long enough to contribute to a background level of PBM.

P7 L30 to P8 L1: As indicated in a previous comment, it is recommended to test this assumption by analyzing the GOM measurements for the presence of a daily cycle in summer. Are there any tracers available, indicating the air was coming from the free troposphere (low rH, low aerosol concentration,. . .)? If not, at least a statement should be made that the trajectory analysis showed that high GOM concentrations were linked to free tropospheric air origin (was not stated before). Answer: About GOM coming from the free troposphere. It is most likely that GOM partly or entirely stem from the free troposphere or the stratosphere, as is suggested by airborne measurements. However, the result from this work only suggests that a great part of the GOM measured at the Råö was formed in the atmosphere as a result of oxidation of GEM. Exactly where these reactions take place could not be studied in lack of complementary data. The text has been changed; accordingly, see my answers above regarding GOM.

P8 L1-2: The meaning of this sentence is not clear. Do the authors mean that these are the first measurements in Northern Europe giving evidence that GOM is formed in the free troposphere? Did ground based and airborne measurements in other regions observed a similar behaviour of GOM (e. g. at other (high elevated) GMOS sites, or studies on vertical distribution of GOM by Lyman and Jaffe 2012, Nature Geosci., 5, 114–117, 2012, and Brooks et al., 2014; Atmosphere, 5, 557–574)? Answer: It should be: Evidence for atmospheric formation and origin of GOM through ground based measurement is presented. When the statement was done in the original manuscript we were not aware of that the same observation also has been done in Nevada, USA, Weiss-Penzias et al. 2009. Hence, that work is now mentioned in the revised text.

Table 1: On page 5 Line 30 it is written that 50% of the GOM measurements were below the detection limit. How did the authors consider measurements below the detection limit for the median/average calculation? Please explain in the text. For Waldhof the authors give only the median and for Aucgencort Moss and Harwell only the average concentration. Would it be possible to calculate median and average for all stations? Answer: How the calculation of median and average values were done will be explained in the text. We have no access to the data from the other sites.

Technical comments: P1 L8: "to the" instead of "tot the" P1 L9: "were" instead of "was" P2 L13-14: Framework Program (FP7) P3 L5: were instead of was P3 L6: please define acronym CVAFS P3 L12: upstream instead of up streams P4 L31: "Six such back trajectories were calculated. . ." maybe better "Six back trajec- C6 tory ensembles were calculated. . ." P5 L4: acronym VBL needs to be defined P6 L9: Fig. 3a should be Fig 3b P6 L17: maybe better write "in air from the south east sector" instead of "in the south east sector" P6 L18: maybe better write "in the air from south east" instead of "in the south east" P12 Figure 1a: I'm not sure it is necessary to show the 1130/1135 flow chart, as it is a standard instrument and the flow chart is given in the manual. If the authors want to show the flow chart, at least a link to the source of the graph is needed. HgP should be PBM, because in the text only PBM is used. P12 Figure 1b: To have a more professional look, the 3d effect with round reflecting edges should be avoided. More space between a) and b) is needed. P14 Fig 4: It is recommended to avoid the color changeover in the individual columns. Instead two different colors should be used to differ between background air and polluted air. The labeling at the x-axis in a), b), and c) is not correct: Januari and Februari should be January and February. If the y-scale in Fig. a) starts at 1 ng m-3, the seasonality would be better to see.

References

Pacyna, E. G., Pacyna, J. M., Steenhuisen, F., and Wilson, S. 2006. Global anthropogenic mercury emission inventory for 2000, Atmos. Environ., 40, 4048–4063.

[Figure]

Pacyna J. M., Travnikov O., Simone F. D., Hedgecock I. M., Sundseth K., Pacyna E. G., Steenhuisen F., Pirrone N., Munthe J., Kindbom K. 2016. Current and future levels of mercury atmospheric pollution on global scale. Atmos. Chem. Phys. Discuss., doi:10.5194/acp-2016-370, 2016. Manuscript under review for journal Atmos. Chem. Phys. Published: 25 May 2016.

Streets, D. G., Zhang, Q., and Wu, Y. 2009. Projections of Global Mercury Emissions in 2050, Environ. Sci. Technol., 43, 2983–2988.

Garland J. A. and Derwent R. G. 1979. Destruction at the ground and the diurnal cycle of concentration of ozone and other gases. Quart. J. R. Met. SOC., 105, pp. 169-183.

Zhang J. and Rao S. T. 1999. The Role of Vertical Mixing in the Temporal Evolution of Ground-Level Ozone Concentrations. J. Appl. Meteorol., vol. 38, 1674-1691.

Coyle M., Smitha R. I., Stedman J.R., Weston K. J., Fowler D. 2002. Quantifying the spatial distribution of surface ozone concentration in the UK. Atm. Environ. 36, 1013-1024.

Weiss-Penzias P., Gustin M. S., Lyman S. N. Observations of speciated atmospheric mercury at three sites in Nevada: Evidence for a free tropospheric source of reactive gaseous mercury. Journal of Geophysical Research - Atmospheres, 07/2009, Vol. 114, Issue D14.

Please also note the supplement to this comment:
http://www.atmos-chem-phys-discuss.net/acp-2016-526/acp-2016-526-AC1-supplement.pdf

---

## Author Comment (AC2) · 4 Oct 2016

Answer to Referee #2

Airborne mercury species at the Råö background monitoring site in Sweden: Distribution of mercury as an effect of long range transport Author(s): I. Wängberg , M. G. Nerentorp Mastromonaco, J. Munthe, and K. Gårdfeldt

MS No.: acp-2016-526 MS Type: Research article Special Issue: Global Mercury Observation System – Atmosphere (GMOS-A)

The authors thank the referee for the help with improving the manuscript. Best regards, Ingvar Wängberg

[Figure]

Comments from referee #2

Excellent paper. Could be published with minor corrections suggested below.

A: -requests for improvements:

Draw a figure (simple box model) to illustrate your argument that the free tropospheric GOM is a likely source of "excess" GOM observed at Rao. (line 10-15, p7)

Answer: The arguments regarding tropospheric GOM has been updated and includes now a lot of more information than previously.

Attempt to compare with the La Seyne sur Mer data presented in Marusczak et al. (2014) Answer: The TGM measurements at La Seyne-sur-Mer covers partly the same time periods as the measurements in Northern Europe (Table 1), but the values are influenced by local mercury sources and therefore not directly comparable.

B: - suggestions for improvement Page 1 Title : Airborne mercury species at the Swedish Rao monitoring site : their distributions are affected by long-range transport.

Abstract : Within the EU-funded GMOS project, . . .mid-May

line 11 : remote location south-east background, free tropospheric air

Introduction Line 26 : particulate-bound mercury Page2 Line1 : chain, which occurs frequently in marine and freshwater ecosystems Line 10: bedrock, and their contributions to. . . are estimated. . . Line 15: Tekran speciation unit was used. . . Line 24: the detail and amount of comparison given in the text does not warrant the amout of "teasing" performed in the introduction. Line 30 GMOS master Line 31: considered a real background site: please provide references or additional information.

Page 3 Line 1: give percentiles/extrema/ standard deviations associated with average meteorological parameters. Line 15: were obtained Line 17: every four hours, three-hour average PBM and GOM values are obtained, together. . .. Line 27: to ensure that all oxidized Line 30: quantified by the . . .

Page4 Line 1: laboratory-built Line 6: Once the blanks are at the appropriate level, PBM and GOM were always detected... Line 11: do reference the GMOS SOP or link to the web-site.

Page 5 Line 5: could be associated with each. Line 24: probably because the Waldhof Line 30 : or GOM. Close to...

Page 6 Line4: simplify sentence Line 11: mercury sources in Poland, Romania, .. Line 12: electricity and domestic heating, but also Line 25: and 5.71 ng mˆ-3 Line 31: influenced by . . .

Page 7 Line 7: bromine-driven photolytic oxidation (Donohoue et al., 2006) Line 8: formed at a slow rate Line 17: south-east... the air sampled at Rao has . . . Line 19: domestic heating Line 24 At Rao, the airborne... Line 29: normally short atmospheric... of GOM, one . . . Line 31, GOM are not likely...

Page 11: Table 1: do separate better the upper table (Median) and lower part (Average)

Page 14 Fig 4: use segmented line plots rather than bars. It would allow to plot means and medians on the same figure. If bars MUST be used, avoid gradient fills. January and February are misspelled.

Please also note the supplement to this comment: http://www.atmos-chem-phys-discuss.net/acp-2016-526/acp-2016-526-AC2-supplement.pdf

---

## Author Response (AR1)

**List of changes in the revised manuscript acp-1016-526**

**Abstract:**

Abstract has been updated with more information.

5

**Chapter 1 Introduction:**

Introduction is now updated with more information on the present work and with more references according to recommendations from Referee #1

10 Chapter 2.1 Sampling site:

Now contains more information on the Råö measurement site according to Referee #1 and #2.

**Chapter 2.2 Methods:**

Chapters 2.2.1 and 2.3 has been shortened according to recommendations from Referee #1

**15**

**Chapter 3.3 Background and polluted:**

Is revised with a lot of more information regarding observations regarding especially GOM as a respond to comments by Referee #1.

**20**

**Airborne mercury species at the Råö background monitoring site in Sweden: Distribution of mercury as an effect of long range transportationtransport**

Ingvar Wängberg1, Michelle G. Nerentorp Mastromonaco2, John Munthe1, Katarina Gårdfeldt2

[revised manuscript text omitted]

**5 Tables**

| Measurement     | CEM/TCM                                | COM                | DDM                | Doriod                                          | Deference            |
|-----------------|----------------------------------------|--------------------|--------------------|-------------------------------------------------|----------------------|
| sites           | median                                 | median             | median             | renou                                           | Kelerence            |
|                 | ng m -3                     | pg m -3 | pg m -3 |                                                 |                      |
| Råö             | 1.41                                   | 0.23               | 2.21               | 2012 - 2015                                     | This work            |
| Mace Head*      | 1. <del>5048</del>              | -                  | -                  | 2011
2013 2012 -
2015              | Weigelt et al., 2015 |
| Waldhof         | 1.61                                   | 1.0                | 6.30               | 2009 - 2011                                     | Weigelt et al., 2013 |
|                 | Average                                | Average            | Average            |                                                 |                      |
|                 | ng m -3                     | pg m -3 | pg m -3 |                                                 |                      |
| Råö             | $1.42\pm0.20$                          | $0.80 \pm 1.6$     | $3.6 \pm 4.5$      | 2012 - 2015                                     | This work            |
| Mace Head*      | 1.4 948 ±
0. <del>2014</del> | -                  | -                  | <del>2011 -</del>
2013 2012 -
2015 | Weigelt et al., 2015 |
| Aucgencort Moss | $1.40 \pm 0.19$                        | $0.43 \pm 1.7$     | 3.1 ± 5.3          | 2009 - 2011                                     | Kentisbeer, 2014     |
| Harwell*        | $1.45\pm0.24$                          | -                  | -                  | 2012 - 2013                                     | Kentisbeer, 2015     |

Table 1. Median and average mercury concentrations measured at some Northern European sites

\*Total gaseous mercury measured with Tekran instruments, calculated for the same periods as in this work

10

Figure 1. a) a schematic overview of the Tekran mercury speciation system 1130/35 and the 2537 mercury analyser, b) picture of the 5 instrument operating at the Råö station.